# Spiking GS: Towards High-Accuracy and Low-Cost Surface Reconstruction via Spiking Neuron-based Gaussian Splatting

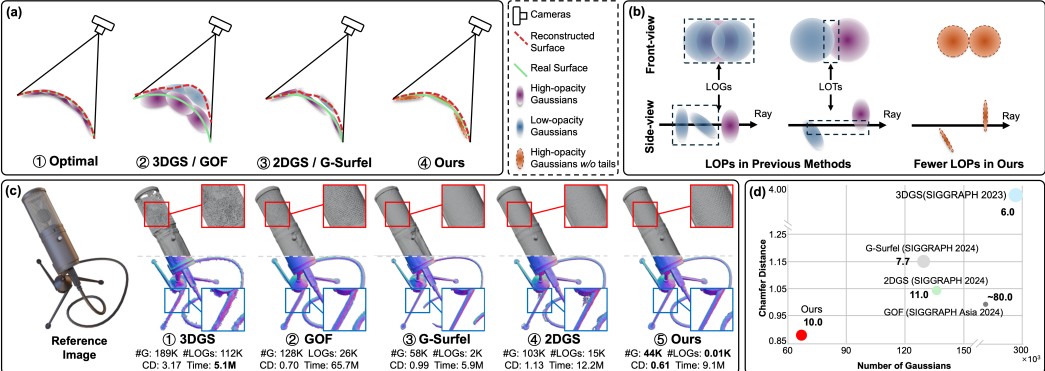

Figure 1: **Figure (a)** intuitively displays the side view of the optimized Gaussian around the surface of interest under different situations: ① optimal (according to (Dai et al., 2024; Guédon & Lepetit, 2024)), by ② 3DGS/GOF, ③ 2DGS/G-Surfel, and ④ ours. The bias of the reconstructed surface can be observed by the distance between the red and green lines. Our Gaussian distribution is better optimized because our approach suppresses the integration of *low-opacity parts (LOPs)* for view rendering. As depicted in **Figure (b)**, LOPs mainly consist of Gaussians with general low-opacity (LOGs, with opacity below a given threshold), and low-opacity Gaussians' tails (LOTs, with Gaussian representation function value below a given threshold) found in every 3D Gaussians. From top to bottom, the first (second) row shows the front (side) view of the Gaussians around the surface of interest. In **Figure (c)**, we show a visual comparison of the extracted mesh for the Mic object from the NeRF-Synthetic dataset (Mildenhall et al., 2021). We additionally compare the Gaussians' number (#G), LOGs' number (#LOGs), Chamfer Distance (CD), and optimization time between our and previous methods. In **Figure (d)**, we show a 2D plot of different methods' averaged statics on the NeRF-Synthetic dataset (Mildenhall et al., 2021) in terms of CD, #G, and training time visualized by the size of the circles, bigger circles indicate faster speed; numbers around circles shows the specific time cost in minutes.

## Abstract

3D Gaussian Splatting is capable of reconstructing 3D scenes in minutes. Despite recent advances in improving surface reconstruction accuracy, the reconstructed results still exhibit bias and suffer from inefficiency in storage and training. This paper provides a different observation on the cause of the inefficiency and the reconstruction bias, which is attributed to the integration of the low-opacity parts (LOPs) of the generated Gaussians. We show that LOPs consist of Gaussians with overall low-opacity (LOGs) and the low-opacity tails (LOTs) of Gaussians. We propose Spiking GS to reduce such two types of LOPs by integrating spiking neurons into the Gaussian Splatting pipeline. Specifically, we introduce global and local full-precision integrate-and-fire spiking neurons to the opacity and representation function of flattened 3D Gaussians, respectively. Furthermore, we enhance the density control strategy with spiking neurons' thresholds and an new criterion on the scale of Gaussians. Our method can represent

more accurate reconstructed surfaces at a lower cost. The code is available at
`https://anonymous.4open.science/r/SpikingGS-D721`.

# 1 INTRODUCTION

3D surface reconstruction from multiview RGB images is a challenging task in computer graphics and vision (Turkulainen et al., 2024; Verbin et al., 2022). Methods based on Neural Radiance Fields (NeRF) (Mildenhall et al., 2021; Guo et al., 2022; Martin-Brualla et al., 2021; Niemeyer et al., 2022; Wang et al., 2021) can extract geometry information from an implicit representation but require extensive computational time for training, limiting their application scenarios. In contrast, 3D Gaussian Splatting (3DGS) (Kerbl et al., 2023) is exceptionally fast in 3D reconstruction, which is gaining more and more attention.

In particular, 3DGS-based methods integrate the opacity of Gaussians for view rendering and surface reconstruction. In the optimal scenario (Dai et al., 2024; Guédon & Lepetit, 2024), only a few *high-opacity* Gaussian tightly clustered around the surfaces contribute to integration, as shown in Fig. 1 (a) ①. However, a suboptimal situation often arises, where Gaussians with low- and high-opacity are distributed around the surfaces, yielding the emergence of excessive *low-opacity parts* (LOPs) in the 3DGS integration (Fig. 1 (a) ②-③ and Fig. 1 (b)). Failing to suppress the emergence of LOPs results in excessive 3D Gaussians, hindering training and storage efficiency. Besides, LOPs also negatively influence surface reconstruction, leading to surface bias, as shown in Fig. 1 (a) ②-③. Most of the previous 3DGS-based surface reconstruction methods overlook the effects of LOPs, but instead focus on improving the reconstructed surface's quality through extra regularization (Dai et al., 2024), enhancing mesh extraction method (Yu et al., 2024), and new Gaussian primitives (flattened 3D Gaussians, Chen et al. 2023; Dai et al. 2024; Huang et al. 2024). Despite the effectiveness of these methods, there is still considerable potential to improve the accuracy and efficiency of reconstruction, as shown in Fig. 1 (c) ①-④.

In this paper, we show a detailed analysis (Sec. 3.2) about above observations in Fig. 1 (a) and Fig. 1 (c), based on which we further identify two primary sources of LOPs: *low-opacity Gaussians* (LOGs) and the *low-opacity tails* (LOTs) of Gaussians. To handle LOGs and LOTs, we introduce a global full-precision integrate-and-fire (FIF) spiking neuron on the opacities of Gaussians and a local FIF spiking neuron considering the Gaussian representation itself, respectively. The former uses a learnable threshold to raise the lower bound of the opacity value of the scene, which effectively reduces the number of LOGs; the latter introduces the discontinuity in the Gaussian representation function, which reduces the contribution of LOTs in 3DGS integration. By suppressing the integration of LOPs for view rendering, we can adaptively regulate LOPs, facilitating robust convergence towards the optimal scenario for various scenes. Although Spiking GS excels in surface reconstruction, it may overly remove Gaussians in occluded or sparsely observed regions, which slightly harms the reconstruction accuracy. To avoid this, we introduce the scale-based clone to compensate for the missing Gaussians in these regions. Moreover, we implement several regularization losses to further enhance method's performance. Extensive experiments show that Spiking GS achieves comparable or superior 3D reconstruction performance on mainstream datasets with much fewer Gaussians and significantly outperforms other methods in semi-transparent scenarios. Note that Spiking GS does not rely on any priors from pre-trained geometry (depth or normal) estimation models. A glimpse of Spiking GS's performance is shown in Fig. 1 (d). In summary, we make the following contributions:

- We analyze the formation of LOPs, which consisting of LOGs and LOTs, are prevalent in the generated results of previous methods. We further show that the integration of excessive LOPs bring the bias of surface reconstruction and the high-cost of the optimiztion.

- We introduce a global FIF spiking neuron on the opacity of Gaussians and a local FIF spiking neuron on each Gaussian representation function, which reduces the number of LOPs and suppresses the integration of LOPs for view rendering.

- We propose optimized training strategies, including the scale-based clone and several regularization losses to improve reconstruction accuracy. Without priors from pre-trained geometry estimation models, the proposed Spiking GS achieves higher reconstruction accuracy with fewer Gaussians for low storage occupation and high optimization efficiency.

## 2   RELATED WORK

**Surface Reconstruction with implicit representation.** Neural Radiance Field (NeRF, Mildenhall et al. 2021) has explored a novel way to reconstruct 3D surface based on a neural radiance field. However, such reconstruction suffers from significant artifacts due to a lack of constraints on geometry (Verbin et al., 2022; Wang et al., 2021; Yariv et al., 2021; Fu et al., 2022; Wang et al., 2022; 2023) and defects of MLP in representing real-world scene (Liao et al., 2024). To improve the reconstruction quality, some of the subsequent works focus on modifying the MLP-based representation by either integrating with Signed Distance Functions (SDFs), a more explicit surface representation (Wang et al., 2021; Yariv et al., 2021; Fu et al., 2022; Wang et al., 2022; 2023), or introducing discontinuities into the density field (Liao et al., 2024). Other works adopt strong priors to regularize the reconstruction, such as priors from depth (Deng et al., 2022; Yu et al., 2022) and MVS (Zhang et al., 2021; Zuo & Deng, 2022). Nevertheless, these methods still require prolonged time for optimizing the neural radiance fields. In contrast, our method is developed on the fast 3DGS method to achieve more accurate surface reconstruction with lower cost.

**Surface Reconstruction with 3D Gaussians.** Although 3D Gaussian Splatting (Kerbl et al., 2023) achieves high fidelity on novel view synthesis, it struggles to reconstruct accurate surface due to the difficulty of aligning primitives with the actual surface (Dai et al., 2024) and the inconsistency between mesh extraction and view rendering (Yu et al., 2024). To address the issues, Chen et al. 2024b; Guédon & Lepetit 2024 utilize a regularization to flatten the 3D Gaussian ellipsoids into flat shapes, while Dai et al. 2024; Huang et al. 2024 directly use flattened 3D Gaussians as primitives. Others redesigned the opacity modeling of Gaussians (Lyu et al., 2024; Yu et al., 2024), improving the details of the reconstructed surfaces. However, these approaches still face challenges in the accuracy of surface reconstruction in scenes including objects with complex materials (*i.e*, semi-transparent). The proposed Spiking GS identifies a widely overlooked correlation between the reconstruction bias and LOPs, achieving considerable improvement on the reconstruction quality and efficiency by reducing the number of LOPs.

**Efficient Geometry Representation of 3D Gaussians.** The use of a large number of Gaussians to accurately represent scenes often results in inefficient training and storage. To address this, recent efforts have focused on utilizing compact feature encodings and pruning strategies to improve efficiency in Gaussian representations (Chen et al., 2024c; Lee et al., 2024; Lu et al., 2024). However, these methods sacrifice surface reconstruction accuracy. Other works (Dai et al., 2024; Guédon & Lepetit, 2024) eliminate the occurrence of LOGs or introduce opacity-based regularization terms to reduce the extra Gaussians. However, they suffer from limited representation capabilities, particularly in scenes with semi-transparent objects, or limited effect on the occurrence of Gaussians. Although our method shares a similar target of reducing the number of Gaussians for improved efficiency, we specifically focus on minimizing the number of LOPs in integration. As a result, we not only effectively achieve the target but also maintain the representation capabilities for accurate surface reconstruction in various scenes.

**Spiking Neural in Surface Reconstruction.** Spiking Neural Networks (SNNs) have received extensive attention recently in many fields due to their energy efficiency (Kim et al., 2022; Ren et al., 2024). It has been utilized in processing video or event signals (Paredes-Vallés & De Croon, 2021; Zhu et al., 2022), classification tasks (Li et al., 2022; Zhang et al., 2022), and adversarial attacks (Ding et al., 2022; Kundu et al., 2021; Liang et al., 2022; Zhou et al., 2021). Recently, Liao et al. (2024) introduce the spiking neuron into the NeRF-based neural surface reconstruction for better performance in representing discontinuities of the scene. The proposed spiking GS is also equipped with the spiking neurons but for its efficiency in removing the LOPs in 3DGS. Experiments show that it significantly improves the performance of 3DGS-based surface reconstruction.

## 3   PRELIMINARIES AND ANALYSIS

In this section, we present preliminaries in Spiking GS, including flattened Gaussian splatting and spiking neurons. We then conduct an analysis of LOPs, which is the core of Spiking GS.

## 3.1 PRELIMINARY

**Flattened Gaussian Splatting**. 3DGS (Kerbl et al., 2023) use a set of 3D Gaussian primitives $\mathcal{G}$ to model the real world scene, each of which is parameterized by an opacity $\alpha \in [0, 1]$, central point's position $\mathbf{p} \in \mathbb{R}^{3 \times 1}$, scale vector $\mathbf{S} \in \mathbb{R}^{3 \times 1}$, and the rotation matrix $\mathbf{R} \in \mathbb{R}^{3 \times 3}$ parameterized by a quaternion $r$. In this paper, we apply flattened 3D Gaussians (Dai et al., 2024) as the 3D primitive due to its ability to reconstruct richer surface details than other methods. The distribution of flattened 3D Gaussian is defined by:

$$\mathcal{G}(\mathbf{x}) = e^{-\frac{1}{2}(\mathbf{x}-\mathbf{p})^\top \Sigma^{-1}(\mathbf{x}-\mathbf{p})}, \tag{1}$$

where x is the 3D coordinate, $\Sigma$ is the covariance matrix expressed as $\mathbf{RSS}^\top \mathbf{R}^\top$, $\mathbf{S}$ is set as $[\mathbf{S}^x, \mathbf{S}^y, 0]^\top$ (Dai et al., 2024). In the rendering process, flattened 3D Gaussians in the world coordinates are transformed into the camera coordinates using a world-to-camera transformation matrix $\mathbf{W}$ and projected onto the image plane using an affine approximation matrix $\mathbf{J}$ (Zwicker et al., 2001):

$$\Sigma' = \mathbf{JW} \Sigma \mathbf{W}^\top \mathbf{J}^\top, \tag{2}$$

where the third row of $\Sigma$ is skipped for projection. After that, the rendered color of each pixel is calculated via alpha-blending of $N$ ordered Gaussians overlapped on pixel points $\mathbf{u}$:

$$\mathbf{c}(\mathbf{u}) = \sum_{i=0}^{N} T_i \omega_i \mathbf{c}_i, \quad T_i = \prod_{j=0}^{i-1} (1 - \omega_j), \quad \omega_i = \mathcal{G}'_i \alpha_i, \tag{3}$$

where $\mathbf{c}(\mathbf{u})$ is the pixel value at point $\mathbf{u}$, $\mathbf{c}_i$ is the RGB color of Gaussians, $\mathcal{G}'_i(\mathbf{u}) = e^{-\frac{1}{2}(\mathbf{u}-\mathbf{p})^\top \Sigma'^{-1}(\mathbf{u}-\mathbf{p})}$ is the Gaussians projected in 2D, $\alpha_i$ is the Gaussian opacity, $\omega_i$ is the weighted Gaussian opacity. During training, Gaussians' attributes are optimized by a photometric loss, and the number of Gaussians is controlled by an adaptive density control process (Kerbl et al., 2023).

**Spiking Neuron**. We introduce the simplified full-precision integrate-and-fire (FIF) spiking neuron (Li et al., 2022; Liao et al., 2024). The model is described as:

$$O = I \cdot s, \quad s = \begin{cases} 0 & I < \bar{V}, \\ 1 & \text{otherwise,} \end{cases} \tag{4}$$

where $I$ is the input value, $\bar{V}$ is a spiking threshold, $O$ is the output value. To make the spiking neuron differentiable, we use the surrogate gradient (Li et al., 2021; Liao et al., 2024) to calculate the gradient during training as:

$$\frac{\partial O}{\partial I} = s, \quad \frac{\partial O}{\partial \bar{V}} = \lambda I \max(0, \frac{k - |I - \bar{V}|}{k^2}), \tag{5}$$

where, $k$ and $\lambda$ are hyperparameters for surrogate gradients.

## 3.2 ANALYZING LOW-OPACITY PARTS

Upon analyzing the pipelines and generated Gaussians of 3DGS methods, we obtain insights below.

**The suboptimal scenario leads to the emergence of excessive LOPs.** As shown in Eq. (3), the pixel value in a given view is computed by integrating the color and opacity of the nearby projected Gaussians. If we focus on reducing the loss between the rendered and ground truth pixel values $\mathbf{c}(\mathbf{u})$ while overlooking regularizing the contribution of each Gaussian, there exists two possible scenarios for deriving the same pixel value $\mathbf{c}(\mathbf{u})$: the scenario where only a few high-opacity Gaussians are involved, or the scenario where both a few high-opacity Gaussians and excessive low-opacity Gaussians are involved. According to (Dai et al., 2024; Guédon & Lepetit, 2024), the latter is suboptimal. We observe that existing state-of-the-art (SOTA) methods tend to converge to such suboptimal scenario. We validate this by visualizing the representative pattern of the $\omega$ values along the view rays generated by SOTA methods in Fig. 2 (a) and (b), showing that excessive LOPs are introduced by excessively generated Gaussians in the SOTA methods. This is further validated by the statistic results in Fig. 2 (c) and (d).

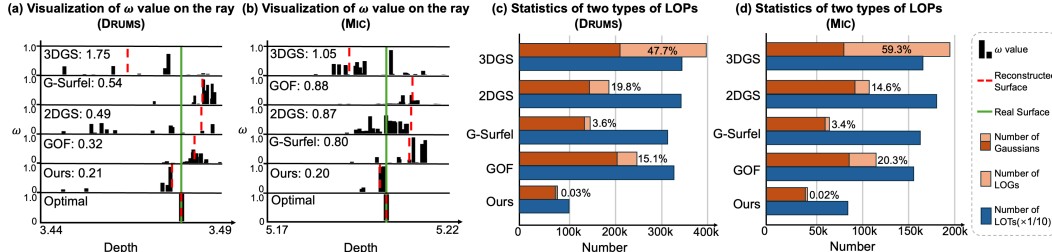

Figure 2: **Visualization of $\omega$ value on a view ray of (a)** DRUMS **and (b)** MIC **from NeRF-Synthetic Dataset** (Mildenhall et al., 2021). We select representative patterns of $\omega$ near the actual surface. The height of each bin represents the $\omega$ value, the red and green lines represent the GT depth and the estimated depth, respectively, which are the same as Fig. 1 (a). We compare five methods (3DGS, 2DGS, G-Surfel, GOF, and ours) and the optimal case. Short bins indicate the presence of LOPs spreading out around the actual surface positions. We also mark the corresponding depth errors (the absolute difference between the ground truth depth and predicted depth) behind the name of each method. **Statistics on two types of LOPs on (c)** DRUMS**, (d)** MIC **from NeRF-Synthetic Dataset**, including the number of Gaussians (total height of the colored bar), the proportion of LOGs (light orange section), and the number of LOTs (blue bar, count as the averaged overlaps of Gaussian's tails with a Gaussian representation function value below 0.1 in multiple views).

**Excessive LOPs bring the bias of surface reconstruction** as extracting surfaces from Gaussians involves similar opacity integration in Eq. (3). Take the widely used TSDF fusion (Huang et al., 2024) as an example. It first computes the depth for every pixel point $\mathbf{u}$ as:

$$t(\mathbf{u}) = \sum_{i=0}^{N} \prod_{j=0}^{i-1} \left(1 - \mathcal{G}'_j \alpha_j\right) \mathcal{G}'_i \alpha_i t_i, \tag{6}$$

where $t_i$ is the depth of the $i$-th 3D Gaussians. According to Eq. (6), the computed depth may be biased by the redundant LOPs. We visualize the estimated bias in Fig. 2 (a) and (b). The error in the depth map will be passed to the extracted mesh since the TSDF fusion method fuses depth maps from different views to generate the final mesh. Such an issue also exists in other advanced surface extraction methods, *e.g.*, the tetrahedral-based surface extraction proposed in (Yu et al., 2024), due to their similar computation method in Eq. (6) for the opacity field derivation. Although we have shown that numerous LOPs bring the bias of surface reconstruction, we would like to emphasize that they are not the only cause, *i.e* fewer LOPs in the integration is one of the necessary conditions towards the optimal Gaussians for 3DGS-based methods. Due to a training process driven by photometric loss instead of direct regularization on geometry, it is possible that the reconstructed surfaces still has a high reconstruction bias even with fewer LOPs. For example, G-Surfel sometimes shows a high depth error but contains fewer LOPs compared to other methods (Fig. 2, (a)).

**LOPs consist of LOGs and LOTs.** As $\omega = \mathcal{G}'\alpha$ (Eq. (3)), either a low $\alpha$ or a low $\mathcal{G}'$ result in a low $\omega$ value, which forms the LOPs in the 3DGS integration. Note that there will be an overlap between these two types of LOPs, as LOTs exist in every Gaussian due to the continuity of Gaussian representation function. Statistics results confirm that both LOGs and LOTs contribute to the 3DGS integration in previous methods (Fig. 2 (c) (d)), leading to bias and inefficiency during the optimization. By utilizing spiking neurons, our method effectively reduces the number of LOPs, significantly improving the accuracy and efficiency in surface reconstruction.

## 4  METHOD

This section presents details of the proposed Spiking GS. We first illustrate how the proposed global and local FIF neurons help reduce the number of LOPs (Sec. 4.1). In Sec. 4.2, we present the training process of Spiking GS to achieve high-accuracy surface reconstruction with a small number of Gaussians. The pipeline is shown in Fig. 3.

Figure 3: **The pipeline of the proposed Spiking GS.** Two types of FIF neurons are applied to the flattened Gaussians. A modified density control process can more adaptively adjust the number of Gaussians. Attributes of the Gaussians are updated by photometric loss $\mathcal{L}_c$, total variance loss $\mathcal{L}_t$, depth distortion loss $\mathcal{L}_d$, normal loss $\mathcal{L}_N$, scale loss $\mathcal{L}_s$, and threshold loss $\mathcal{L}_\alpha$, $\mathcal{L}_{\mathrm{p}}$.

## 4.1 FIF NEURONS IN SPIKING GS

The key of the proposed Spiking GS method is leveraging FIF neurons to reduce the number of LOPs in integration for view rendering and surface reconstruction. Specifically, we employ the global FIF neuron to manage LOGs, and the local FIF neuron to control LOTs. These spiking neurons guide the training process, encouraging Gaussians to converge towards the optimal scenario, where fewer Gaussians, with minimal overlap in their tails, represent the scene (Fig. 1 (a) ④, Fig. 1 (b) right). This brings improved accuracy and efficiency in surface reconstruction.

**Global FIF neuron on opacity**. The global FIF neuron is applied to the Gaussians' opacity $\alpha$. The output of FIF neuron is calculated as:

$$\hat{\alpha} = \begin{cases} 0 & \alpha < \bar{V}^\alpha \\ \alpha & \text{otherwise,} \end{cases} \tag{7}$$

where $\bar{V}^\alpha$ is shared by all Gaussians and optimized during training. In the original method, a key criterion for pruning Gaussians is whether their opacity is below a predefined threshold $\epsilon_\alpha$. Additionally, a reset strategy is employed, setting all Gaussians' opacities to a low value $\epsilon_\alpha^{\mathrm{r}}$, together with pruning to manage the increasing number of Gaussians. Although this approach is effective in reducing floaters near the camera, it is less successful in reducing the number of LOGs around surfaces. By replacing $\epsilon_\alpha$ and $\epsilon_\alpha^{\mathrm{r}}$ with $\bar{V}^\alpha$, Gaussians with $\alpha < \bar{V}^\alpha$, typically corresponding to LOGs, are efficiently removed. In addition, the use of a larger adaptive reset threshold can implicitly guide 3DGS to converge toward the optimal case across various scenes.

**Local FIF neuron on representation function**. The tails of Gaussians (low function value parts) often overlap when they are positioned closely together. This overlap can sometimes contribute to the integration for view rendering and surface reconstruction, resulting in biases and an excessive number of Gaussians. To address this, we "cut off" the tails of the 3D Gaussian representation function to reduce such overlap. This is accomplished by introducing the local FIF (full-precision integrate-and-fire) neuron into the Gaussian representation function, redefined as:

$$\hat{\mathcal{G}}_i(\mathbf{x}) = \begin{cases} 0 & \mathcal{G}_i(\mathbf{x}) < \bar{V}_i^{\mathrm{P}} \\ e^{-\frac{1}{2}(\mathbf{x}-\mathbf{p})^\top \Sigma^{-1}(\mathbf{x}-\mathbf{p})} & \text{otherwise,} \end{cases} \tag{8}$$

where $\bar{V}^{\mathrm{P}}$ is the cut-off threshold set as a learnable parameter. Notably, each Gaussian is assigned a unique $\bar{V}_i^{\mathrm{P}}$, reflecting the fact that different regions in the scene may need varying cut-off thresholds to capture the geometry accurately. For instance, areas near object boundaries might necessitate a larger cut-off threshold to model the discontinuity at the boundary than other parts of the object. We find the local FIF neurons effectively reduce the contribution of low-opacity tails (LOTs), minimizing unwanted overlap and improving the accuracy and efficiency of the 3DGS. In practice, to better integrate $\hat{\mathcal{G}}$ into the rasterization framework, we directly apply the local FIF neuron on the projected 2D Gaussian on the screen, making it act *as if* applying the model on the 3D Gaussian.

## 4.2 OPTIMIZING SPIKING GS

Despite the success in using fewer Gaussians to represent the scene, artifacts may appear on the reconstructed surfaces, as a tighter clustering of Gaussians around the surfaces is required. To address this, additional regularization on the reconstructed surfaces is necessary. Furthermore, we observe that the proposed FIF neurons and the enhanced Gaussian pruning strategy may excessively remove Gaussians in regions with limited constraints (*e.g.*, blind spots or areas captured by only a few views), which negatively impacts reconstruction accuracy. To mitigate this, we introduce a new criterion in the original cloning process, we call it the scale-based clone. The following section summarizes the loss functions implemented and explains the scale-based cloning process.

**Threshold loss.** This is proposed to enhance the effect of local and global FIF neurons. Two regularization terms $\mathcal{L}_\alpha$ and $\mathcal{L}_p$ is introduced to increase $\bar{V}^\alpha$ and $\bar{V}^p$ during training:

$$\mathcal{L}_\alpha = \lambda_\alpha \frac{1}{\bar{V}^\alpha}, \quad \mathcal{L}_p = \lambda_p \frac{1}{\bar{V}^p}, \tag{9}$$

where $\lambda_\alpha = \lambda_p = 2e^{-5}$.

**Scale loss.** As presented in (Fan et al., 2024), large-sized Gaussians tend to magnify minor errors during training and hinder the Gaussians' ability to capture high-frequency geometric details. Motivated by this insight, we propose a simple scale loss to constrain the scale of Gaussians:

$$\mathcal{L}_s = \sum_i \mathcal{R}\left(\max(\mathbf{S}_i^x, \mathbf{S}_i^y)\right), \mathcal{R}(\mathrm{m}) = \begin{cases} 0 & \mathrm{m} < \bar{V}^\theta \\ \mathrm{m} & \text{otherwise,} \end{cases} \tag{10}$$

where $\bar{V}^\theta$ is a hyperparameter that controls the extent of scale regularization.

**Total loss.** Excluding the losses on $\bar{V}^\alpha$, $\bar{V}^p$, and $\mathbf{S}$, we implement losses of previous methods. Except for the photometric loss $\mathcal{L}_c$ following (Kerbl et al., 2023), we also use depth distortion loss $\mathcal{L}_d$, normal loss $\mathcal{L}_N$ following (Huang et al., 2024), and total variance loss $\mathcal{L}_t$ following (Karnieli et al., 2022) to regularize the geometry. To sum up, the total loss $\mathcal{L}$ is calculated as:

$$\mathcal{L} = \mathcal{L}_c + \mathcal{L}_d + \mathcal{L}_N + \lambda_\alpha \mathcal{L}_\alpha + \lambda_p \mathcal{L}_p + \lambda_s \mathcal{L}_s + \lambda_t \mathcal{L}_t, \tag{11}$$

where $\lambda_s$ and $\lambda_t$ are the hyperparameters set as $5e^{-4}$ and 1.0 respectively.

**Scale-based Clone**. The geometry of blind spots or regions observed by limited viewpoints is particularly difficult to reconstruct due to the lack of constraints from viewpoints. Additionally, we find Gaussians in those regions hardly meet the clone criteria of splitting and cloning, which may be overly removed by the enhanced pruning strategy, resulting in deficiency (*e.g.*, undesired holes and pits) in the reconstructed surface. To mitigate this issue, we propose a scale-based clone. Specifically, it additionally clone Gaussians that meet $\mathcal{R}\left(\max(\mathbf{S}_i^x, \mathbf{S}_i^y)\right) \in [\theta - \delta, \theta + \delta]$, where $\delta = \frac{\theta}{200}$. The new clone rule enriches the number of Gaussians in blind spot regions, improves the opportunity for those Gaussians to be guided by view-related loss, compensates for the geometric information deficiency, and improves the reconstruction quality in these regions.

## 5 EXPERIMENTS

Our training framework is developed on 3DGS (Kerbl et al., 2023) with flattened Gaussians (Dai et al., 2024) as primitive and customized CUDA kernels[1].

**Evaluation metrics and datasets.** To evaluate the accuracy of the 3D surface reconstruction, we use the Chamfer distance as the metric (Dai et al., 2024; Huang et al., 2024; Liao et al., 2024). We also report training time and the number of Gaussians (noted as #G) to evaluate the efficiency of our method. We test our method in 8 scenes from NeRF-Synthetic dataset (Mildenhall et al., 2021), 6 scenes from Dex-NeRF dataset (Ichnowski et al., 2021), and 15 scenes from DTU dataset (Jensen et al., 2014). These datasets include both synthetic and real-world scenes.

**Baselines.** We compare our method's performance with one NeRF-based neural surface reconstruction method (Spiking NeRF (Liao et al., 2024), noted as "SpNeRF") and five Gaussian-based

---

[1]For parameters setup and training details, please refer to the appendix.

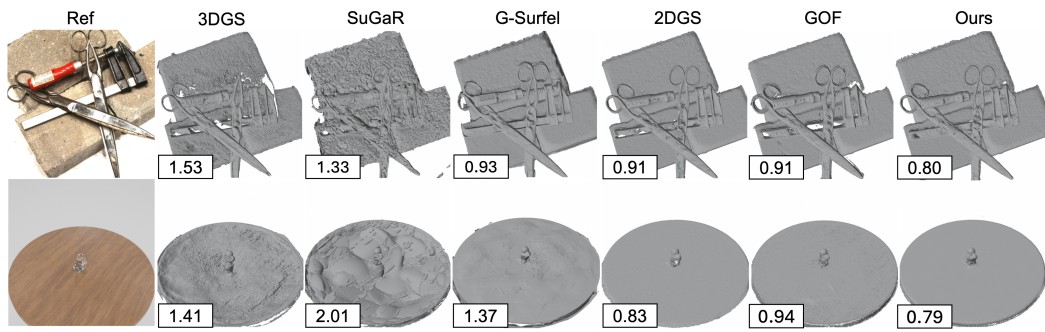

Figure 4: **Qualitative Comparisons of surface reconstruction** performed on DTU (Jensen et al., 2014) and Dex-NeRF datasets (Ichnowski et al., 2021). We show the Chamfer distance in the bottom left corner of the image.

Table 1: Quantitative Comparison on NeRF-Synthetic dataset (Mildenhall et al., 2021). We report the Chamfer distance ($\times 10^{-2}$) for the reconstructed mesh in 8 scenes, the number of Gaussians (#G), and training time.

|  | CHAIR | DRUMS | FICUS | HOTDOG | LEGO | MATERIALS | MIC | SHIP | AVG | #G | Time |
|---|---|---|---|---|---|---|---|---|---|---|---|
| SpNeRF | 0.66 | 2.43 | 0.54 | 0.94 | 0.70 | 1.10 | 0.72 | 1.49 | 1.07 | - | > 12h |
| 3DGS | 2.76 | 3.62 | 8.38 | 5.38 | 2.44 | 2.92 | 3.17 | 2.18 | 3.86 | 293k | **6.0 m** |
| SuGaR | 0.67 | 1.08 | 0.57 | 1.42 | 0.73 | 0.95 | 0.87 | 1.19 | 0.94 | - | ~40 m |
| G-Surfel | 0.75 | 0.78 | 0.56 | 1.77 | 0.93 | 1.08 | 0.99 | 2.35 | 1.15 | 128k | 7.7 m |
| 2DGS | 0.40 | 1.83 | 0.66 | 1.41 | 0.87 | 0.93 | 1.13 | 1.11 | 1.04 | 133k | 11.0 m |
| GOF | 1.23 | 1.33 | 0.59 | 1.18 | 0.72 | 0.72 | 0.70 | 1.35 | 0.98 | 184k | 1.2h |
| Ours | 0.47 | 1.38 | 0.69 | 1.13 | 0.81 | 0.94 | 0.61 | 0.96 | 0.87 | **69k** | 10.0 m |
| - *w/o* glb | 0.46 | 1.48 | 0.74 | 1.12 | 0.88 | 0.98 | 0.64 | 1.02 | 0.92 | 238k | 12.6 m |
| - *w/o* loc | 0.47 | 1.37 | 0.70 | 1.10 | 0.81 | 0.90 | 0.60 | 1.00 | 0.88 | 85k | 10.3 m |

surface reconstruction methods (*i.e*, 3DGS (Kerbl et al., 2023), SuGaR (Guédon & Lepetit, 2024), G-Surfel (Dai et al., 2024), 2DGS (Huang et al., 2024), and GOF (Yu et al., 2024)). To further evaluate the training and storage efficiency, we compare our method with two compression methods (*i.e*, Compact-3DGS (Lee et al., 2024), denoted as "C3DGS", and HAC (Chen et al., 2024c)).

## 5.1 GEOMETRY RECONSTRUCTION

**Quantitative Comparison.** To quantitatively assess the quality of the reconstructed surface, we compared the proposed Spiking GS with other methods on the NeRF-Synthetic (Mildenhall et al., 2021), Dex-NeRF (Ichnowski et al., 2021), and DTU (Jensen et al., 2014) datasets. The results are shown in Tab. 1-3. respectively. According to the results, our method achieves SOTA performance on most datasets (NeRF-Synthetic and Dex-NeRF). We find that the adaptability of the FIF neurons improves the method's robustness, showing stability in various scenes, especially with semi-transparent objects. However, we also notice that the advantage diminishes on the DTU dataset primarily because the viewpoints in the DTU are concentrated on the front side of the object, with few views from the back. This results in limited constraints introduced from the back views, negatively impacting our method.

**Qualitative Comparison.** To evaluate the quality of the reconstructed surfaces, we present the results of all methods for selected objects from two datasets in Fig. 4. While methods like 3DGS and SuGaR exhibit some high-frequency noise on the reconstructed surfaces, our method achieves both smoothness and preservation of intricate surface details. The advantage is obvious on the surface with complex materials (*e.g.*, the reflective surface of the metallic object and the glass object in Fig. 4). We attribute the improvements to the proposed spiking neurons, which adaptively guide Gaussians toward convergence in the optimal scenario.

**Efficiency Comparison.** As shown in Tables 1-3, our method demonstrates a significant reduction in storage (nearly twice fewer Gaussians than 2DGS (Yu et al., 2024) on the NeRF-Synthetic and DTU dataset, and even five times fewer than 2DGS (Yu et al., 2024) on the Dex-NeRF dataset) while maintaining a competitively fast training time. Additionally, we also compare Spiking GS with other

Table 2: Quantitative Comparison on selected scenes of Dex-NeRF dataset (Ichnowski et al., 2021). We report the Chamfer distance ($\times 10^{-2}$), the number of Gaussians (#G), and training time.

| | PIPE | MOUNT | UP | PAWN | TURBINE | SIDE | AVG | #G | Time |
|---|---|---|---|---|---|---|---|---|---|
| SpNeRF | 0.82 | 1.10 | 0.83 | 0.83 | 1.27 | 1.26 | 1.02 | - | > 12h |
| 3DGS | 1.24 | 1.34 | 1.37 | 1.41 | 1.43 | 1.14 | 1.36 | 483k | 11.4 m |
| SuGaR | 3.36 | 1.97 | 0.84 | 2.01 | 0.79 | 3.31 | 2.05 | - | ~ 40 m |
| G-Surfel | 1.41 | 1.38 | 1.29 | 1.37 | 1.30 | 1.44 | 1.37 | 47k | 39.0 m |
| 2DGS | 0.81 | 0.83 | 1.46 | 0.83 | 1.13 | 0.85 | 0.98 | 123k | 9.3 m |
| GOF | 0.90 | 0.86 | 0.75 | 0.94 | 0.80 | 1.53 | 0.96 | 305k | 2h |
| Ours | 0.76 | 0.77 | 1.40 | 0.79 | 0.77 | 0.80 | 0.88 | **24k** | **7.5 m** |
| - *w/o* glb | 0.81 | 0.80 | 1.07 | 0.91 | 1.00 | 1.24 | 0.97 | 195k | 9.7 m |
| - *w/o* loc | 0.79 | 1.32 | 1.02 | 0.83 | 0.90 | 1.53 | 1.07 | 367k | 8.2 m |

Table 3: Quantitative Comparison on 15 scenes from DTU dataset (Jensen et al., 2014). We report the Chamfer distance, the number of Gaussians (#G), and training time.

| Method | 24 | 37 | 40 | 55 | 63 | 65 | 69 | 83 | 97 | 105 | 106 | 110 | 114 | 118 | 122 | Mean | #G | Time |
|---|---|---|---|---|---|---|---|---|---|---|---|---|---|---|---|---|---|---|
| SpNeRF | 0.84 | 1.20 | 1.02 | 0.38 | 1.15 | 0.72 | 0.69 | 1.10 | 1.19 | 0.65 | 0.49 | 1.60 | 0.49 | 0.55 | 0.51 | 0.83 | - | > 12h |
| 3DGS | 2.14 | 1.53 | 2.08 | 1.68 | 3.49 | 2.21 | 1.43 | 2.07 | 2.22 | 1.75 | 1.79 | 2.55 | 1.53 | 1.52 | 1.50 | 1.96 | 542k | 11.2m |
| SuGaR | 1.47 | 1.33 | 1.13 | 0.61 | 2.25 | 1.71 | 1.15 | 1.63 | 1.62 | 1.07 | 0.79 | 2.45 | 0.98 | 0.88 | 0.79 | 1.33 | - | ~ 1h |
| G-Surfel | 0.66 | 0.93 | 0.54 | 0.41 | 1.06 | 1.14 | 0.85 | 1.29 | 1.53 | 0.79 | 0.82 | 1.58 | 0.45 | 0.66 | 0.53 | 0.88 | **166k** | 16.5m |
| 2DGS | 0.48 | 0.91 | 0.39 | 0.39 | 1.01 | 0.83 | 0.81 | 1.36 | 1.27 | 0.76 | 0.70 | 1.40 | 0.40 | 0.76 | 0.52 | 0.80 | 224k | 18.8m |
| GOF | 0.52 | 0.91 | 0.40 | 0.37 | 1.11 | 0.88 | 0.72 | 1.18 | 1.26 | 0.75 | 0.72 | 0.88 | 0.46 | 0.71 | 0.55 | 0.76 | 369k | 1.8h |
| Ours | 0.63 | 0.80 | 0.50 | 0.46 | 1.04 | 1.01 | 0.87 | 1.15 | 1.32 | 0.61 | 0.73 | 1.19 | 0.44 | 0.68 | 0.48 | 0.79 | 171k | **11.1m** |
| - *w/o* glb | 0.69 | 0.86 | 0.68 | 0.48 | 1.17 | 1.27 | 0.92 | 1.19 | 1.35 | 0.63 | 0.76 | 1.21 | 0.50 | 0.76 | 0.54 | 0.87 | 427k | 15.9m |
| - *w/o* loc | 0.73 | 0.81 | 0.68 | 0.49 | 1.04 | 1.02 | 0.93 | 1.22 | 1.31 | 0.60 | 0.77 | 1.28 | 0.45 | 0.69 | 0.49 | 0.83 | 191k | 11.2m |

compression methods (C3DGS (Lee et al., 2024) and HAC (Chen et al., 2024c)) in terms of Chamfer distance, number of Gaussians, and training time. The results are shown in Tab. 4, indicating our method balances the reconstruction accuracy, training efficiency, and storage efficiency. In practice, many factors may affect the method's efficiency. For instance, although reducing the number of Gaussians during training can shorten training time, the additional operations (*e.g.*, regularization computation) required to achieve this may increase time, offsetting the benefits. This explains why our method is faster than C3DGS and HAC, despite sometimes using more Gaussians. Additionally, the number of training epochs plays a critical role. For example, with a pre-trained normal estimation model, G-Surfel requires only 15000 iterations for good performance. In contrast, our method and other 3DGS approaches, lacking such priors, need more iterations.

## 5.2 ABLATION STUDY

To validate the effectiveness of the proposed methods, we conduct an ablation study of the proposed Spiking GS. The results of a comparison between our method and two alternatives, '*w/o* glb' and '*w/o* loc', are shown in Tab.1 [2]. Specifically, '*w/o* glb' removes the global FIF neuron, while '*w/o* loc' excludes the local FIF neuron. The results show that omitting either the global or local FIF neurons negatively impacts both the accuracy and efficiency of surface reconstruction. The two types of FIF neurons play distinct roles in our method for reducing LOPs, resulting in different effects. For instance, the global FIF neuron is primarily effective in reducing the number of LOGs to improve reconstruction quality and efficiency. Particularly, we find method with global FIF neuron can better capture the details on the surface, validated by a quality comparison between the full model and '*w/o* glb' in Fig. 5, column 1-3. In contrast, local FIF neurons decrease LOTs to minimize the overlap of Gaussians, resulting in more separated centers of Gaussians, as evidenced by the visualized Gaussian center's distribution in Fig. 5, columns 5-6. Local FIF neurons are particularly effective in specific scenes, such as scenes with flat surfaces (the round table in Fig. 4, column 2) where tail overlap is predominant. Incorporating local FIF neurons will be more efficient in these cases to reduce the number of Gaussians, compared to using global FIF neurons.

## 6 CONCLUSION

We present Spiking GS, a novel method for more accurate surface reconstruction at a lower cost. Our method reduces the number of low-opacity parts (LOPs) in the integration process for view

---

[2]More results can be found in the appendix.

Table 4: Performance comparion between C3DGS (Lee et al., 2024), HAC (Chen et al., 2024c), and Spiking GS (ours) on three datasets (Ichnowski et al., 2021; Jensen et al., 2014; Mildenhall et al., 2021). We report the averaged Chamfer distance (CD), the number of Gaussians (#G), and reconstruction time. **Bold numbers** indicate the best results.

| Dataset | NeRF-Synthetic | | | Dex-NeRF | | | DTU | | |
|---------|------|------|------|------|------|------|------|------|------|
| Metric | CD | #G | Time | CD | #G | Time | CD | #G | Time |
| C3DGS | 4.59 | 118k | **10m** | 2.41 | 62k | 12m | 2.56 | 329k | 22m |
| HAC | 3.88 | **56k** | 13m | 4.72 | 114k | 20m | 4.33 | **61k** | 17m |
| Ours | **0.87** | 69k | **10m** | **0.88** | **24k** | **8m** | **0.79** | 171k | **11m** |

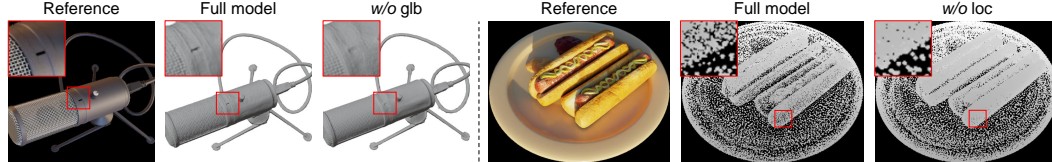

Figure 5: The qualitative comparison between the Spiking GS and its alternatives. From left to right: columns 1-3 are comparisons of reconstructed mesh between the full model and 'w/o glb', and columns 4-6 are comparisons of Gaussians' center points projected on a specific view between the full model and 'w/o loc'.

rendering, which we have identified as one of the causes of inefficiency and reconstruction bias in Gaussian-based methods. Spiking GS incorporates global and local FIF neurons into the flattened Gaussians to address two types of LOPs: *low-opacity Gaussians* (LOGs) and *low-opacity tails* (LOTs) of Gaussians. Additionally, we implement a modified density control method and regularization losses into the training process. As a result, Spiking GS demonstrates robust performance in various datasets, achieving considerable improvements.

**Limitation.** Despite the effectiveness of the proposed Spiking GS, it still has several limitations. First, the reconstructed surface in blind spot regions exhibits strong bias, requiring additional geometric priors. Second, high-frequency details (*e.g.* very thin structures) are challenging to reconstruct due to inherent shortcomings in mesh extraction methods. Addressing these limitations will be regarded as future work.

**Additional Discussion.** We notice several concurrent works (Fan et al., 2024; Chen et al., 2024a) that achieve impressive surface reconstruction accuracy on the DTU dataset. It is important to highlight that LOPs remain prevalent in their generated Gaussians, indicating potential for further refinement. Our method is compatible with additional losses, priors, or regularization, allowing it to improve or maintain surface reconstruction accuracy while significantly reducing the number of Gaussians. Moreover, Spiking GS demonstrates robust performance across various scenes, including challenging semi-transparent ones. In contrast, other methods may be restricted to specific scene types. Experimental comparison and more discussion can be found in the appendix.

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
