# Appendix for "Spiking GS: Towards High-Accuracy and Low-Cost Surface Reconstruction via Spiking Neuron-based Gaussian Splatting"

## A  IMPLEMENTATION DETAILS

Following Huang et al. 2024, we modify the rasterization renderer to output depth and normal maps for regularization. For parameters setup, we follow the basic setup of learning rates in (Kerbl et al., 2023). Particularly, to enhance training stability, we reduce the learning rates of $\bar{V}^\alpha$ and $\bar{V}^p$ from $2e^{-4}$ to 0 during first 300 iterations of every 3000 iterations within the initial 15000 iterations. For mesh extraction during evaluation, we use truncated signed distance fusion (TSDF) to extract meshes from depth maps (Huang et al., 2024). In our implementation, we set the voxel size to 0.004 and the truncation threshold to 0.02. All of our experiments are conducted on a single 24GB NVIDIA RTX3090 GPU.

## B  DETAILS OF REGULARIZATION TERMS

**Depth distortion loss.** Optimizing solely on $\mathcal{L}_c$ (Kerbl et al., 2023) can result in noisy surfaces, so we follow Huang et al. 2024 to introduce depth distortion loss. Depth distortion loss $\mathcal{L}_d$ reduces the depth disparity along the ray, concentrating the Gaussian splats to be closer to each other:

$$\mathcal{L}_d = \sum_{i,j} \lambda_i^d \lambda_j^d |t_i - t_j|, \tag{12}$$

where $\lambda_i^d = \alpha_i' \mathcal{G}_i'(\mathbf{x}) \prod_{j=1}^{i-1} \left(1 - \alpha_j' \mathcal{G}_j'(\mathbf{x})\right)$ is the blending weight of $i$-th Gaussian and $t_i$ is its depth. Since directly using the depth of Gaussian' center $\mathbf{p}$ can introduce errors (Dai et al., 2024), we follow Dai et al. 2024 to use the depth at the ray-Gaussian intersection instead.

**Normal loss.** Normal consistency loss $\mathcal{L}_n$ (Huang et al., 2024) helps the Gaussians to align with the actual surfaces by ensuring consistency between the Gaussians' normal and the surface normal:

$$\mathcal{L}_N = \sum_i \lambda_i^N (1 - \mathbf{n}_i^\top \mathbf{N}), \tag{13}$$

where $\mathbf{n}_i$ denotes the normal of Gaussian and $\mathbf{N}$ is the surface normal estimated with the gradients of the depth maps (Huang et al., 2024). To smooth the estimated surface normal, we further apply a bilateral filter (Elad, 2002) on the depth maps.

**Total variance loss.** Following (Karnieli et al., 2022; Turkulainen et al., 2024), we apply the edge-aware total variance loss on depth maps to smooth the surface representation:

$$\mathcal{L}_t = \sum_{ij} |\partial_x \hat{d}_{ij}| e^{-\|\partial_x \bar{I}_{ij}\|} + |\partial_y \hat{d}_{ij}| e^{-\|\partial_y \bar{I}_{ij}\|}, \tag{14}$$

where $\partial_x$ and $\partial_y$ are the gradients in the horizontal and vertical directions, $\hat{d}_{ij}$ is the estimated depth at pixel $(u_i, v_j)$ on depth maps, and $\bar{I}$ is the average color of ground truth images. This regularization term improves smoothness of depth maps while offers a faster convergence. Note that we apply the non-edge-aware form of Eq. (14) for the Dex-NeRF dataset (Ichnowski et al., 2021) as a further regularization.

Table 6: Ablation study on NeRF-Synthetic dataset (Mildenhall et al., 2021). We compare the proposed Spiking GS (Full) with its 5 alternatives (*i.e*, 'w/o $\mathcal{L}_\alpha$', 'w/o $\mathcal{L}_p$', 'w/o $\mathcal{L}_s$', 'w/ GP', 'w/o c').

|  | Full | w/o $\mathcal{L}_\alpha$ | w/o $\mathcal{L}_p$ | w/o $\mathcal{L}_s$ | w/ GP | w/o c |
|---|---|---|---|---|---|---|
| CD↓ | 0.87 | 0.91 | 0.89 | 1.03 | 1.07 | 0.92 |
| #G↓ | 69k | 245k | 74k | 55k | 68k | 64k |

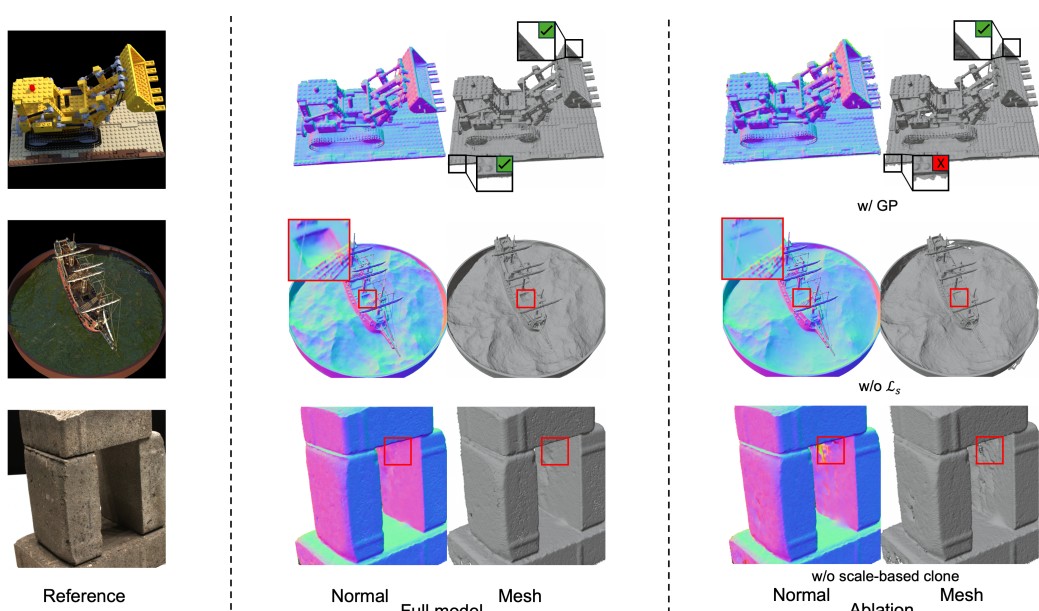

Figure 7: The qualitative comparison between the Spiking GS and its alternatives. From top to down: ablation on the necessity of a local FIF neuron on each Gaussian representation function, ablation on scale loss $\mathcal{L}_s$, and ablation on scale-based clone.

## C  ADDITIONAL ABLATION STUDIES

**Necessity of a local FIF neuron on each Gaussian representation function** is validated by a comparison with the alternative, 'w/ GP'. Specifically, 'w/ GP' uses a global threshold shared by all Gaussians. As shown in Tab. 6, a shared threshold will dramatically downgrade the accuracy of the reconstructed surface. Through a qualitative comparison (Fig. 7), we identify that the downgrading in accuracy is caused by unnecessary extension of surface (*e.g.*, the base board of LEGO). The analysis above indicates the necessity of a isolate FIF neuron on each Gaussian representation function to fit the geometry at different locations.

**Effectiveness of proposed loss** is assessed by comparison between Spiking GS and its three alternatives (*i.e*, 'w/o $\mathcal{L}_\alpha$', 'w/o $\mathcal{L}_p$', and 'w/o $\mathcal{L}_s$'). As can be seen on Tab. 6, both the number of Gaussians and Chamfer distance increase without $\mathcal{L}_\alpha$ and $\mathcal{L}_p$, proving the loss on $\bar{V}^\alpha$ and $\bar{V}^p$ boosting the effect of FIF neurons. Additionally, a quantitative (Tab. 6) and qualitative (Fig. 7) comparison between the full model and 'w/o $\mathcal{L}_s$' demonstrate the effect of $\mathcal{L}_s$ in improving surface's details.

**Effectiveness of scale-based clone** is proved by a comparison between the full model and alternative without the scale-based clone strategy (w/o c). As shown in the Fig. 7, artifacts (*e.g.*, holes and pits) are caused by the blind spots regions (*i.e*, the inner side of stone pillars) with less opportunity to be cloned in original density control process. The proposed scale-based clone compensates for insufficient Gaussian points in those regions, resulting in higher surface reconstruction accuracy. A quantitative result without such strategy, shown in Tab. 6, further validates our analysis.

Table 7: Additional quantitative Comparison on NeRF-Synthetic (Mildenhall et al., 2021) dataset between Spiking GS, PGSR (Chen et al., 2024), and its alternative (S-PGSR, integrated with our method). We show the Chamfer distance ($\times 10^{-2}$) for the reconstructed mesh in 8 scenes, as well as the number of Gaussians used for geometry reconstruction (#G) and training time.

| | CHAIR | DRUMS | FICUS | HOTDOG | LEGO | MATERIALS | MIC | SHIP | AVG | #G | Time |
|---|---|---|---|---|---|---|---|---|---|---|---|
| Ours | 0.47 | 1.38 | 0.69 | 1.13 | 0.81 | 0.94 | 0.61 | 0.96 | 0.87 | **69k** | **10.0 m** |
| PGSR | **0.38** | 1.17 | 0.54 | 1.06 | **0.74** | 1.41 | **0.66** | 0.73 | 0.84 | 205k | 24.3 m |
| S-PGSR | **0.38** | **1.15** | **0.52** | **1.01** | **0.74** | **1.30** | **0.66** | **0.70** | **0.81** | 77k | 23.5 m |

# D ADDITIONAL DISCUSSION

Although some concurrent works (Fan et al., 2024; Chen et al., 2024) reconstruct accurate geometry on the DTU dataset (Jensen et al., 2014), we find that these methods still suffer from issues caused by excessive LOPs and poor reconstruction results on some challenging scenes (*e.g.*, semi-transparent objects from the Dex-NeRF dataset), where both methods fail to generate reasonable results.

**Discussion about TrimGS (Fan et al., 2024)**. Fan et al. 2024 introduced a novel density control strategy to trim inaccurate Gaussians based on a pre-trained Gaussian model. However, it tends to overly split and generate numerous Gaussians if the pre-trained model contains excessively large LOPs. According to our experiment, the number of Gaussians of the trimmed 2DGS (Fan et al., 2023; Huang et al., 2024) could exceed ten million in NeRF-Synthetic (Mildenhall et al., 2021) and Dex-NeRF (Ichnowski et al., 2021) datasets, which severely undermine training efficiency and consume a significant amount of VRAMs.

**Discussion about PGSR (Fan et al., 2024)**.Chen et al. 2024 utilized a multiview geometry consistency prior constraint to regularize the reconstructed surface, exhibiting strong performance in smooth surface reconstruction. Nevertheless, they overlooked the prevalence of LOPs and the issues associated with. Our method can be implemented into their pipeline. Specifically, we integrate our FIF spiking neurons into their method. Through a quantitative comparison among our method, the original PGSR, and PGSR with spiking neurons (S-PGSR) on the NeRF-Synthetic dataset in Tab. 7, we observe an improvement in reconstruction accuracy and efficiency in surface reconstruction, further validating the need to reduce the number of LOPs.

# E ADDITIONAL RESULTS

We shown additional qualitative comparisons result on Dex-NeRF dataset (Ichnowski et al., 2021), NeRF-Synthetic dataset (Mildenhall et al., 2021), and DTU dataset (Jensen et al., 2014).

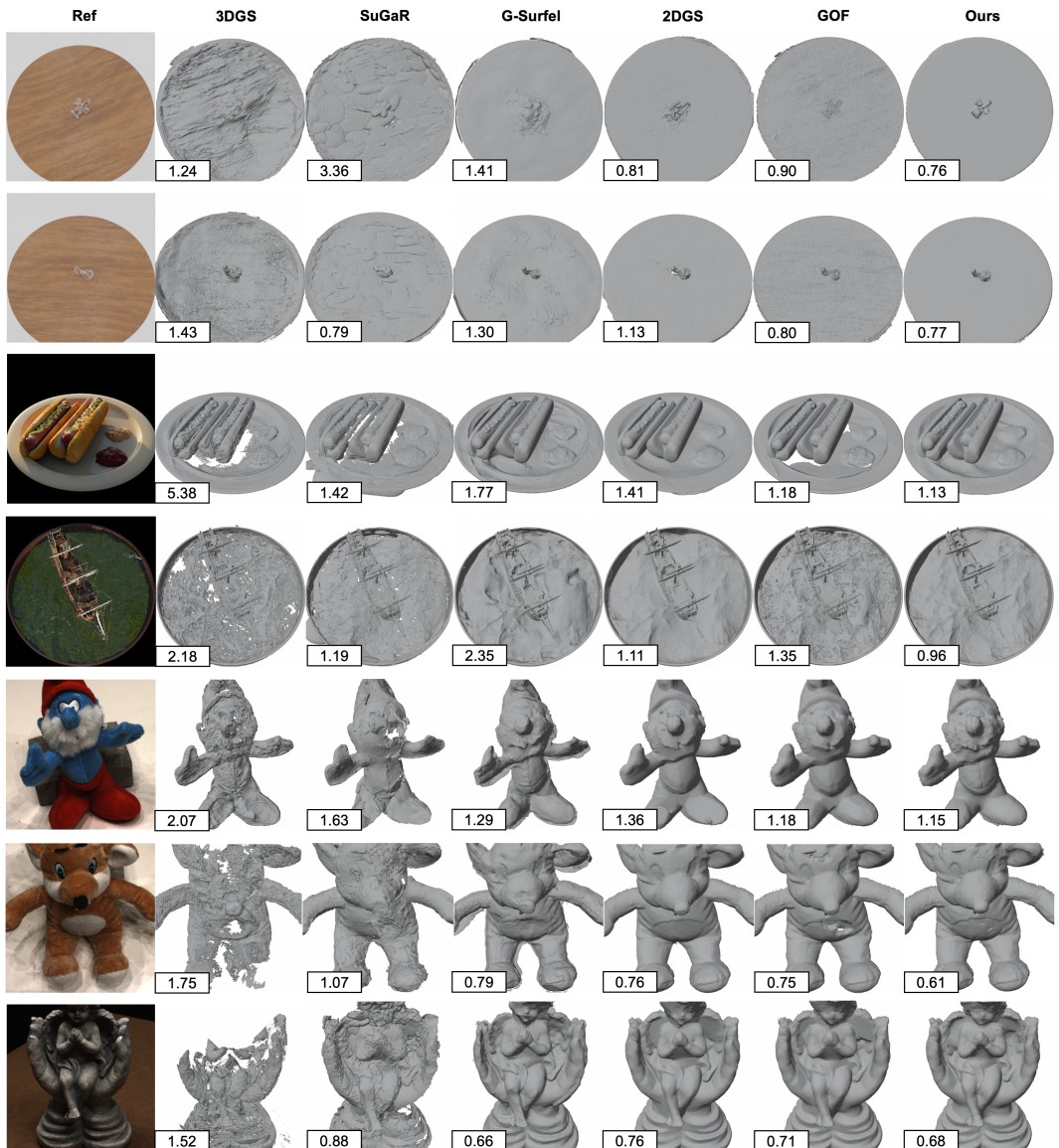

Figure 8: **Additional qualitative comparisons of surface reconstruction** performed on Dex-NeRF (Ichnowski et al., 2021), NeRF-Synthetic (Mildenhall et al., 2021), and DTU (Jensen et al., 2014) datasets. We show the Chamfer distance in the bottom left corner of the image.