# OpenReview forum: "Spiking GS: Towards High-Accuracy and Low-Cost Surface Reconstruction via Spiking Neuron-based Gaussian Splatting"
_ICLR.cc/2025/Conference — ICLR 2025 Conference Withdrawn Submission_

### Official Review · Reviewer_t4dh · 2024-10-28

**Soundness:** 2
**Presentation:** 2
**Contribution:** 2
**Rating:** 5
**Confidence:** 2

**Summary:**

This paper points out that low opacity parts (LOPs) in 3D Gaussian Splatting negatively affect the performance and efficiency of the reconstruction, and introduces global and local full-precision IF spiking neurons to reduce the LOPs and thus improve the performance and efficiency. In addition, the authors propose scale-based clone to further improve the reconstruction quality. Experiments on several datasets show that the proposed method has better performance and efficiency than the existing methods.

**Strengths:**

1. This paper points out that LOPs in 3D Gaussian splicing can negatively affect the performance and efficiency of reconstruction, and further decomposes them into LOGs and LOTs.
2. The paper introduces global and local full-precision IF spiking neurons to reduce LOPs, thereby improving performance and efficiency.
3. The authors further optimize the proposed method including threshold loss, scale loss, and scale-based clone to improve the reconstruction quality.
4. The authors experimentally demonstrated the effectiveness and performance advantages of the proposed method.

**Weaknesses:**

1. The full-precision IF neurons used by the authors (Eqs. 7 and 8) appear to contain only step functions with learnable thresholds, which is significantly different from typical spiking neurons with multiple time steps. This contributed to the limited innovation of this paper.
2. The paper lacks ablation studies for comparison with baseline, making it difficult to adequately understand the effects of each module. For example, the comparison of the proposed method with w/o glb and w/o loc is shown in Table 1~3, but it does not show how much of the improvement over the baseline is achieved by using only glb and loc.
3. It is recommended that the authors reorganize the structure of the paper, e.g., more detailed ablation studies such as loss functions should be placed in the paper rather than in the Supplementary Material to help understand these contributions.

**Questions:**

Please see weakness.

---

> ### Author Response · Authors · 2024-11-15
>
> Thank you for your comments. We will reorganize the paper in the improved version. To address your concern, we would like to make the following clarifications:
>
> ## Q1. The full-precision IF neurons contributed to the limited innovation of this paper.
>
> We **never claim** that a new spiking neurons model is one of our contributions. Instead, we introduce it in the preliminary section (Section 3.1) with proper citations. Our key contribution is to use the spiking neurons to address LOPs in Gaussian Splatting methods. We feel confused about this misunderstanding.
>
> ## Q2. Lacks ablation studies for comparison with baseline.
>
> Ablation study on only glb and loc results will be merged into an improved version. We appreciate your suggestions to help the reader better understand our paper. Our method can **simply integrate with concurrent SOTA works [PGSR] to achieve stable improvement in accuracy and efficiency**, as shown in the table below.
> | Dataset Metric   | NeRF-Synthetic   CD ↓ | NeRF-Synthetic  #G ↓ | NeRF-Synthetic  Time ↓ | DTU  CD ↓ | DTU  #G ↓ | DTU  Time ↓ |
> |-----------|--------------------------|--------------------------|----------------------------|---------------|---------------|-----------------|
> | PGSR      | 0.84                     | 205k                     | 24.3m                      | 0.55          | 165k          | 27.6m           |
> | PGSR  with spiking Neurons  | 0.81                     | 77k                      | 23.5m                      | 0.56          | 88k           | 24.1m           |

---

### Official Review · Reviewer_BgaK · 2024-10-29

**Soundness:** 2
**Presentation:** 3
**Contribution:** 2
**Rating:** 5
**Confidence:** 5

**Summary:**

The authors propose a 3D Gaussian splatting method based on spiking neurons to address inefficiency and reconstruction bias in traditional 3D Gaussian splatting. They claim that the bias is largely due to the integration of low-opacity parts (LOPs) in the generated Gaussians. To counter this, they introduce global and local full-precision IF spiking neurons to manage the opacity and representation function of flattened 3D Gaussians, respectively. Additionally, they enhance the density control strategy by leveraging spiking neuron thresholds and implementing a new criterion based on Gaussian scales.

**Strengths:**

* The paper provides a theoretical analysis of the LOPs issue in 3D Gaussian splatting, accompanied by well-executed visualizations.

* The approach balances the presence of LOPs by employing global and local FIF spiking neurons, which effectively suppress the integration of LOPs during view rendering.

* Optimized training strategies are introduced, including a scale-based cloning technique and various regularization losses, aimed at enhancing reconstruction accuracy.

* This method achieves high-quality 3D reconstruction with a reduced parameter count, making it both efficient and effective.

**Weaknesses:**

* From Equations (7) and (8), the described FIF method seems to follow a spiking neuron approach where activation occurs only when a threshold is exceeded, maintaining traditional 3D Gaussian splatting in other cases. If this interpretation is correct, could this approach be seen as using spiking neuron properties to perform a type of "pruning" operation, thereby reducing parameter count?

* In terms of reconstruction results, the proposed method appears to produce smoother outputs than traditional 3D Gaussian splatting, potentially at the cost of high-frequency details. Can this loss of texture or spatial information be understood as a side effect of the “pruning” or “truncation” mechanism mentioned earlier?

* The comparative results presented so far are limited, making it difficult to fully assess the advantages of this method.

* Should the symbols in Equations (4) and (5) represent vectors? It may be helpful for the authors to review these expressions. Additionally, if the equations involve integration, the meaning of the ⋅\cdot⋅ symbol needs further clarification, as it currently does not seem fully aligned with the integrate-and-fire concept of spiking neurons.

* While the illustrations are visually appealing, the font size is rather small, and the detailed content in each image can be challenging to discern.

**Questions:**

* Certain terms lack clear definitions, such as the meaning of "full-precision" in line 84. Additionally, in the acronym LOGs, what does the "G" stand for?

* To my knowledge, there are already many methods combining neuromorphic cameras with 3D Gaussian splatting, where the cameras generate spike/event data. Could the spiking neuron approach in this paper be readily adapted for use with such spike/event data?

* Why was color reconstruction not used? 3D Gaussian splatting has been widely applied for RGB data reconstruction, so it seems feasible to directly compare colored 3D reconstructions, especially with results across multiple views.

---

> ### Author Response · Authors · 2024-11-15
>
> Thank you for your comments. We will make sure that all the mentioned symbolic minors (e.g., equations (4) and (5)) will be fixed, and the concerned terms, such as full precision and LOGs (Low Opacity Gaussians), will be clearly defined. To address your concern, we would like to make the following clarifications:
>
> ## Q1: Could this approach be seen as using spiking neuron properties to perform a type of "pruning" operation, thereby reducing parameter count?
>
> The proposed spiking neurons help conduct a more adaptive pruning operation. Specifically, the spiking neurons set the Gaussian’s opacity to zero when it is lower than a learnable threshold, which means that Gaussians with zero opacity may exist in the scene and make no contribution to the mesh extraction and rendering. These Gaussians will be removed by pruning operation. The biggest advantage of using spiking GS is that **it makes pruning differentiable and learnable**, through optimizing the learnable threshold during training, which not only reduces LOGs, but also brings performance advantage.
>
> ## Q2: The proposed method appears to produce smoother outputs than traditional 3D Gaussian splatting, potentially at the cost of high-frequency details
>
> We claim that our method **preserves the details**, although we produce smoother surfaces. This can be observed by the qualitative comparison in Figures 4 and 8 (e.g., the details of the semi-transparent objects are well-preserved, and other details like the water’s texture in the fourth row of Figure 8.)
>
> ## Q3: The comparative results presented so far are limited, making it difficult to fully assess the advantages of this method.
>
> We have presented all the quantitative results and part of the representative qualitative comparison results in both the main paper and supplementary materials. We have also conducted an ablation study on the main contribution of this paper. We would appreciate **more concrete suggestions** on this issue.
>
> ## Q4: Could the spiking neuron approach in this paper be readily adapted for use with such spike/event data?
>
> We believe necessary transformation is required as we are targeting different tasks.
>
> ## Q5: Why was color reconstruction not used?
>
> We didn’t include the evaluation of novel view synthesis quality in the submitted version since we would like to emphasize more on our contribution to accurate surface reconstruction. Evaluations on novel view synthesis quality are shown in the table below in terms of PSNR, SSIM, and LPIPS. Note that despite our PSNR being slightly lower than 3DGS, we would like to emphasize our advantages in surface reconstruction and efficiency.
> We believe that less competitive results for view synthesizing are reasonable. Because our method introduces several loss functions targeting better geometry **rather than better rendering quality**.
>
> | Dataset Metric   | NeRF-Synthetic PSNR ↑ | NeRF-Synthetic SSIM ↑ | NeRF-Synthetic LPIPS ↓ | Dex-NeRF PSNR ↑ | Dex-NeRF SSIM ↑ | Dex-NeRF LPIPS ↓ |
> |-----------|------------------------|------------------------|------------------------|-----------------|-----------------|------------------|
> | 3DGS      | 31.05                 | 0.959                 | 0.051                 | 21.84          | 0.904          | 0.287           |
> | G-Surfel  | 31.10                 | 0.954                 | 0.050                 | 30.77          | 0.953          | 0.123           |
> | 2DGS      | 30.28                 | 0.956                 | 0.054                 | 22.19          | 0.886          | 0.280           |
> | GOF       | 33.78                 | 0.969                 | 0.031                 | 20.01          | 0.913          | 0.241           |
> | Ours      | 30.23                 | 0.954                 | 0.050                 | 30.59          | 0.953          | 0.111           |

---

### Official Review · Reviewer_XdWY · 2024-10-30

**Soundness:** 2
**Presentation:** 3
**Contribution:** 2
**Rating:** 5
**Confidence:** 4

**Summary:**

This article proposes a 3DGS method based on pulse activation for scene reconstruction. Especially, the analysis of LOPS, LOGS, and LOTS in this article is very exciting and clear, which will help future researchers to carry out further work. Some innovative narratives that are too simplistic can easily lead to ambiguity. However, the author emphasizes that this article focuses on large-scale scene reconstruction and has not been experienced in experiments.I think this article still needs to enrich and modify the narrative logic, and improve the experimental part.

**Strengths:**

This article provides a detailed analysis of the crucial opacity issue in the 3DGS reconstruction series. The analysis of the impact of LOPS, LOGS, and LOTS is very thorough, which assists the subsequent description and experimental development. The article has clear logic and combines the latest third-generation pulse neurons and 3DGS to achieve a new efficient reconstruction architecture.

**Weaknesses:**

There are some issues with this article: (1) Based on the description in the article that the FIF neuron seems to be a creative work in other works, this article adds global and local limitations. Moreover, the author did not provide a detailed explanation of the differences from the original FIF neurons in the article. Based on the author's brief description, I believe this tends to be a simple call to FIF neurons rather than an innovation. (2) In addition, the author did not clearly state the specific details of the scale based clone, which appears to be very lacking in innovation. The experimental part also did not verify this cloning method. (3) The experimental section lacks visualization of large-scale scenes and does not focus on the rendering effect of new views. (4) Verification of missing loss components. The author emphasizes in the introduction that this article did not use geometric constraints, but the loss definition contains a large amount of prior supervision, which is contradictory. Meanwhile, the impact of adding these losses on performance has not been verified.

**Questions:**

(1) Please describe the essential differences between FIF neurons used in Spiking GS and FIF neurons;
(2) Please describe the guidance of global FIF and local FIF on the convergence process of GS;
(3) Please describe the role of mixing multiple geometric priors for supervision;
(4) The description of Scale based Clone is too brief. Please add it;
(5) Please explain the design logic of the experimental section;
(6) The visualization section shows that there is not much difference between Spike GS and GOF. Please provide more comprehensive comparison results for reference.

---

> ### Author Response · Authors · 2024-11-15
>
> Thank you for your comments. To address your concern, we would like to make the following clarifications:
> ## Q1: Innovation about the FIF neurons.
> Yes, our FIF model is a simple call of the classical FIF neurons. However, we **never claim** that a new spiking neurons model is one of our contributions. Instead, we introduce it in the **preliminary section (Section 3.1) with proper citations**. Our key contribution is to use the spiking neurons to address LOPs in Gaussian Splatting methods. We feel confused about this misunderstanding.
>
> ## Q2: More details about the scale-based clone.
> We apologize for the typos in line 360, which may have confused our scale-based cloning strategy. The corrected criterion we added to the original Gaussian clone process should be $max(S_i^x, S_i^y) \in [\bar{V}^{\theta} - \sigma, \bar{V}^{\theta} + \sigma] $, without function $R(\cdot)$. This means that Gaussians with the largest scale values falling within this range will be cloned.
>
> The **motivation** behind this strategy is that, in blind-spot regions, constraints from the views are weak, resulting in a small view-space positional gradient. Consequently, Gaussians in these regions are not correctly cloned. We chose $\theta$, which is also the hyperparameter for our scale loss (line. 340), because the scale loss is the only view-invariant loss that will become dominant in these regions. Therefore, points with a scale around this value are highly possible to be the points in blind-spot regions. We have quantitatively and qualitatively verified the effectiveness of this strategy in supplementary material, Table 6. and Figure 7.
>
> ## Q3: Visualization of large-scale scenes.
> We **didn’t state** that we are working on large-scale scenes anywhere in the main paper or supplementary material. Our work, as well as other related works (3DGS, G-Surfel, 2DGS), focuses more on a relatively small-scale scene compared to some works that focus on large-scale scene reconstruction. The datasets are particularly chosen to evaluate the geometry reconstruction accuracy as the ground truth meshes are provided. We feel confused about this misunderstanding.
>
> ## Q4: Evaluation of the rendering effect of new views.
> Evaluations on novel view synthetics quality are shown in the table below in terms of PSNR, SSIM, and LPIPS. Note that despite our PSNR being slightly lower than 3DGS, we would like to emphasize **our advantages in surface reconstruction and efficiency**. We believe such results are **expectable** since our method introduces several loss functions targeting better geometry rather than better rendering quality.
>
> We didn’t include the above analysis in the submitted version since we would like to emphasize our contribution to accurate and efficient surface reconstruction.
>
> | Dataset Metric   | NeRF-Synthetic PSNR ↑ | NeRF-Synthetic SSIM ↑ | NeRF-Synthetic LPIPS ↓ | Dex-NeRF PSNR ↑ | Dex-NeRF SSIM ↑ | Dex-NeRF LPIPS ↓ |
> |-----------|------------------------|------------------------|------------------------|-----------------|-----------------|------------------|
> | 3DGS      | 31.05                 | 0.959                 | 0.051                 | 21.84          | 0.904          | 0.287           |
> | G-Surfel  | 31.10                 | 0.954                 | 0.050                 | 30.77          | 0.953          | 0.123           |
> | 2DGS      | 30.28                 | 0.956                 | 0.054                 | 22.19          | 0.886          | 0.280           |
> | GOF       | 33.78                 | 0.969                 | 0.031                 | 20.01          | 0.913          | 0.241           |
> | Ours      | 30.23                 | 0.954                 | 0.050                 | 30.59          | 0.953          | 0.111           |
>
> ## Q5: Confusion about geometric constraints.
> We apologize for the misunderstanding of the statement “Spiking GS does not rely on any priors from pre-trained geometry (depth or normal) estimation models” in line 96. Our primary aim is to highlight the effectiveness of Spiking GS in achieving accurate surface reconstruction without relying on pre-trained models like G-Surfel. Nevertheless, empirical geometric constraints, such as depth distortion loss and total variance loss, are also crucial for precision in surface reconstruction, and including these does not diminish our contributions. While previous methods focus on refining priors to improve accuracy, they overlook the inherent deficiencies in 3DGS representations. The proposed Spiking GS improve the accuracy and efficiency of surface reconstruction in a different, novel perspective, by identifying and addressing the inherent deficiencies of 3DGS (i.e., the LOPs).

---

> ### Author Response · Authors · 2024-11-15
>
> ## Q6: Ablation study on other loss components.
> We add additional quantitative comparisons on the NeRF-Synthetic dataset for three losses. Specifically, w/o $L_d$ stands for without depth distortion loss (lines 26-35 in supplementary material), w/o $L_N$ stands for without normal loss, and w/o $L_t$ stands for total variance loss. We observe different extents of performance degradation, as shown in the table below. However, this doesn’t hinder the main contribution of Spiking GS since we provide a unique perspective to improve the accuracy and efficiency of surface reconstruction in Gaussian Splatting. Our method can simply **integrate with the concurrent SOTA method [PGSR] to achieve further improved results**, as shown in the table below.
> |  | Full Model | w/o $L_N$ | w/o $L_d$ | w/o $L_t$ |
> |--------|------------|---------|---------|---------|
> | CD ↓   | 0.87       | 1.94    | 0.88    | 0.88    |
>
> | Dataset Metric   | NeRF-Synthetic   CD ↓ | NeRF-Synthetic  #G ↓ | NeRF-Synthetic  Time ↓ | DTU  CD ↓ | DTU  #G ↓ | DTU  Time ↓ |
> |-----------|--------------------------|--------------------------|----------------------------|---------------|---------------|-----------------|
> | PGSR      | 0.84                     | 205k                     | 24.3m                      | 0.55          | 165k          | 27.6m           |
> | PGSR  with spiking Neurons  | 0.81                     | 77k                      | 23.5m                      | 0.56          | 88k           | 24.1m           |
>
> ## Q7: The essential differences between FIF neurons used in Spiking GS and FIF neurons
> There is **no essential difference** in modeling between FIF neurons in Spiking GS and the mainstream FIF neurons. However, we set time steps as 1 in our model, implementing a simplified version of FIF neurons tailored for Gaussian Splatting, which differs from most FIF neurons. The idea of using spiking can be found in our response to reviewer MMX4 (Q1).
>
>
>
> ## Q8: The guidance of global FIF and local FIF on the convergence process of GS
> The guidance from global FIF and local FIF during training is **implicit**. As explained in lines 305 and 308, the global FIF neurons raise the lower bound of the Gaussians’ opacity in the scene to remove LOGs; the local FIF removes the low function value part to mitigate the influence of LOTs. Both help the GS method converge with relatively high opacity and less number of Gaussians.
>
> ## Q9: The role of mixing multiple geometric priors for supervision
> These geometric priors help further improve the reconstruction accuracy. Their effect is shown in Q6. As shown in Table in Q6, our method can be **integrated into the concurrent SOTA method [PGSR]** with different geometric priors. We emphasize that these losses do not hinder the contribution of this paper, as shown in Q8.
>
> ## Q10: The design logic of the experimental section
> Our experiments section is organized in the following logic:
> 1. At the beginning of Section 5, we introduce evaluation metrics and baselines targeting both surface reconstruction and Gaussian compression.
> 2. In Section 5.1, we provide quantitative, qualitative, and efficiency comparisons for surface reconstruction to demonstrate the effectiveness of the proposed method on surface reconstruction. And we further provide efficiency comparisons with Gaussian compression methods, as shown in Table 4.
> 3. In Section 5.2, we conduct an ablation study on two proposed spiking neurons to illustrate their contribution further. We put more ablation studies in the supplementary material due to the page limit.
>
> ## Q11: The visualization section shows that there is not much difference between Spike GS and GOF
> We will add a zoom-in window to magnify the details for easier comparison. It is clear that the Spike GS produces smoother geometry than GOF without losing many details. We also observe noticeable advantages in reconstructed surfaces for semi-transparent objects.

---

### Official Review · Reviewer_MMX4 · 2024-11-01

**Soundness:** 3
**Presentation:** 3
**Contribution:** 3
**Rating:** 6
**Confidence:** 3

**Summary:**

This work innovatively integrates spiking neurons into 3D Gaussian Splatting, effectively addressing issues related to low-opacity parts. The paper presents solid experimental validation, demonstrating advantages in both efficiency and effectiveness.

**Strengths:**

1. They provide insightful analysis of the low-opacity parts (LOPs) problem, categorizing it into LOGs and LOTs, and propose global and local FIF neurons as solutions, representing an innovative approach.
2. The proposed method demonstrates effectiveness across several benchmarks, achieving state-of-the-art performance while significantly reducing computational overhead.
3. The writing is clear and accessible, with coherent logical progression throughout, and supported by comprehensive experimental validation.

**Weaknesses:**

1. Insufficient theoretical explanation for the effectiveness of spiking neurons in addressing LOPs.
2. Limited discussion on the impact of parameter selection on performance in section 4.2.

**Questions:**

I would appreciate a more detailed explanation of the precise mechanisms through which the global and local FIF neurons address LOGs and LOTs, respectively. The specific interaction between these components and their respective roles in mitigating opacity-related challenges merits further elaboration.

---

> ### Author Response · Authors · 2024-11-15
>
> Thank you for your comments. To address your concern, we would like to make the following clarifications:
> ## Q1: A theoretical explanation for the effectiveness of spiking neurons in addressing LOPs and the detailed mechanisms.
> The discontinuity (Eq. 4) and differentiability (Eq. 5) properties of spiking neurons align well with our requirements for addressing LOPs. To be Specific,
> - As shown in Section 3.2, we provide **thorough evidence** that LOPs (we further propose that LOPs consist of LOGs and LOTs) are prevalent in current Gaussian Splatting methods, with a detailed explanation for LOPs’ occurrence. Theoretically, this is due to an inherent ambiguity in the optimizing 3D Gaussians since it is the weighted summation of multiple Gaussian opacity and colors that are constrained by the photometric loss (Eq. 3), not individual Gaussians.
> - While Gaussian Splatting has developed pruning and opacity-resetting strategies to remove LOGs, these approaches rely on **pre-set thresholds**, which may not effectively adapt to diverse scene types and often struggle to reduce LOGs that overlap tightly near the surface (Fig 1 (a) and Fig 2). Additionally, current methods overlook the potential negative impact of LOTs.
> - In contrast, our method leverages the discontinuity of spiking neurons to directly address LOPs and the differentiability of these neurons with **learnable thresholds** to enhance adaptability across different scene types. Specifically, global FIF neurons set Gaussians' opacity to zero when their values fall below a threshold (Eq. 7), allowing these LOGs to be pruned more effectively. This strategy increases the overall opacity of scenes to mitigate LOGs, as our threshold is higher than in previous methods.
> - Moreover, local FIF spiking neurons eliminate low-function-value parts (i.e., the tails of the Gaussian function) by setting them to zero when below a threshold (Eq. 8). This reduces the impact of LOTs in rendering and mesh extraction. Due to the differentiability of the thresholds used in FIF neurons, these values are optimized through losses, **enabling more adaptive control over Gaussian number and opacity compared to fixed-threshold methods**.
>
> ## Q2: Discussion on the impact of parameter selection on performance.
> We empirically use three different hyperparameters $\bar{V}^\theta$ for these three datasets, $\bar{V}^\theta = 0.02$ for scenes in NeRF-Synthetic, $\bar{V}^\theta = 1$ for scenes in Dex-NeRF, and $\bar{V}^\theta = 0.01$ for scenes in DTU.
> The settings are based on our observations:
> - for scenes in Dex-NeRF, we observe smoother surfaces, so a higher parameter is preferred.
> - for scenes in NeRF-Synthetic and DTU, we observe more intricate surfaces, so a lower parameter is preferred.
>
> As suggested in line 339, the hyperparameters for $\lambda_\alpha$ and $\lambda_p$ are selected for a general case.

---

### Author Response · Authors · 2024-11-15

On behalf of the authors of the paper titled “Spiking GS: Towards High-Accuracy and Low-Cost Surface Reconstruction via Spiking Neuron-based Gaussian Splatting,” we have decided to withdraw the paper.

Before doing so, we will reply to the reviewers’ comments to provide clarification. We sincerely appreciate the time and effort each reviewer has devoted to evaluating our work, and we are especially grateful for feedback that identified minor ambiguities and suggested valuable areas for further discussion, which helps us improve the current version. However, we also feel confused about some feedback resulting from misunderstandings that are difficult to reconcile with our intended contributions.

---

### Note · Authors · 2024-11-15

I have read and agree with the venue's withdrawal policy on behalf of myself and my co-authors.